# Clinical Outcomes of Monolithic Zirconia Crowns on Posterior Natural Abutments Performed by Final Year Dental Medicine Students: A Prospective Study with a 5-Year Follow-Up

**DOI:** 10.3390/ijerph20042943

**Published:** 2023-02-08

**Authors:** Giuseppe Barile, Saverio Capodiferro, Giovannino Muci, Antonio Carnevale, Giovanni Albanese, Biagio Rapone, Massimo Corsalini

**Affiliations:** 1Department of Interdisciplinary Medicine, University of Bari “Aldo Moro”, 70124 Bari, Italy; 2Dental School, University of Bari “Aldo Moro”, 70124 Bari, Italy

**Keywords:** monolithic zirconia, fixed dental prosthesis, CDA score, dental prosthesis, dental students

## Abstract

The conventional metal–ceramic is still considered the gold standard in fixed prosthetics especially in terms of longevity. Among alternative materials used, Monolithic Zirconia has shown the capability to reconcile excellent biomechanical properties with acceptable aesthetic performance and to overcome several inconveniences related to veneer restorations. This study aims to clinically evaluate Monolithic Zirconia prosthetic crowns on natural abutments in the posterior sectors, performed by final-year dental medicine students (undoubtedly with less experience in the management of such material) by the standardized California Dental Association score system evaluation, to better understand the viability of Monolithic Zirconia. This prospective study was carried out at the Dental School of the University of Bari “Aldo Moro”, Italy. Prosthetic rehabilitation included single crowns or a short pontic prosthesis with maximum one intermediate. Final-year dental students performed tooth reduction under the supervision of three expert tutors. The California Dental Association systematics (based on color, surface, anatomical shape, and marginal integrity) were adopted to evaluate the prosthetic maintenance status over time. Annual follow-up visits were re-evaluated by the same parameters each year. Univariate logistic regression analysis was performed to evaluate outcomes and the Kaplan–Meier plot to report survival. The sample consists of 40 crowns performed on 31 patients, 15 males (48.4%) and 16 females (51.6%) with an average age of 59.3 years. The clinical cases subjected to experimental study were found to be “Excellent” (1a/2a/3a/4a) in 34 cases (85%), “Acceptable” in 4 cases (10%), and “To be re-done” in 2 cases (failures) (5%). Our conclusive data support the predictability of Monolithic Zirconia restorations on natural posterior abutments at a long-term follow-up of five years, even when performed by less-experienced clinicians.

## 1. Introduction

The main goal of any prosthetic rehabilitation is the aesthetic and functional restoration by a medical device able to replace a missing part of the body. There is limited availability of extant restorative materials that simultaneously exhibit both excellent aesthetic and biomechanical properties [1]. As such, despite the great technological progress of the last two decades, the conventional metal–ceramic is still considered the gold standard in fixed prosthetics, especially thanks to its longevity and relatively easy management [2].

However, the increasing patient demand for aesthetic excellence has prompted researchers to focus on improving the optical and mechanical properties of all-ceramic materials. This has led to increased predictability and popularity, with these all-ceramic materials capable of replacing metal–ceramics in almost all clinical applications [1,3].

The shift of clinicians toward metal-free restorations is occurring worldwide [4]. In the context of Advanced Ceramics, these metal-free restorations have stood out as having the best ability to meet both biomechanical (in terms of flexural strength and toughness) and aesthetic requirements [5].

More than 15 years ago, Partially Stabilized Zirconia (PSZ), also known as “conventional Zirconia”, was commercialized. It is characterized by a high light refractive index and considerable opacity. In fact, these characteristics have always represented an aesthetic limitation, so this material was mainly used for substructure, in place of the conventional metal, resulting in a more invasive approach and a higher risk of technical complications [5,6].

Nowadays, Zirconia is commercially available in the form of Tetragonal Zirconia Polycrystals (TZP), and is commonly stabilized with yttria to improve its flexural strength and toughness values [7]. Over the last few years, manufacturing companies have sought to improve the aesthetic properties of Monolithic Zirconia to make it more translucent, reaching good biomechanical and aesthetic performances with the third generation material, known as “Fully Stabilized Zirconia ” or “Cubic Zirconia” [8]. Nevertheless, most authors agree that Monolithic Zirconia is suitable only for posterior tooth rehabilitation because of its poor optical properties [9,10].

Additional advantages of the Monolithic Zirconia are its handling simplicity, the reduced biological implication in terms of tooth reduction, the low cost and the comparatively fast production in the dental laboratory [11]. Few studies have considered the low learning curve required to reach excellence in its use and exploit all of its potential, especially in no/less-experienced hands [12,13]; moreover, most studies were in vitro or with short term follow-up.

The current prospective study aimed to test the viability of Monolithic Zirconia crowns on posterior teeth performed by final year dental students, considered as less experienced clinicians operating under tutor supervision, by clinical outcome evaluation at a 5-year follow up and correlating crown failures with patient and framework characteristics variables, considering the null hypothesis as the lack of correlations.

## 2. Materials and Methods

This prospective study was conducted at the University Hospital “Policlinico” of Bari, Italy, in accordance with the Declaration of Helsinki. The protocol was approved by the Ethical Committee of University Hospital “Policlinico di Bari” (N. Prot. 0069684). All patients were treated during 2017 at the Dental School of the University of Bari “Aldo Moro” and had previously accepted and signed the relevant informed consent.

Prostheses were all single crowns (SC) or multiple-fixed dental prosthesis (FDP) with a maximum of three elements with a single intermediate, all performed by final-year students of Dental Medicine under the supervision of 3 experienced tutors. In no case were extension cantilevers performed.

### 2.1. Patient Sampling

The following inclusion criteria were adopted for this study:Monolithic Zirconia crowns on natural abutments;Posterior sector: premolar or molar regions;Good periodontal health;Stable occlusion;Maximum one intermediate pontic;Absence of systemic diseases as a contraindication to prosthetic treatment.

Patients with the following characteristics were excluded:Poor oral hygiene;High risk of caries;Severe periodontal disease;Parafunctions.

### 2.2. Operating Protocol: Workflow of Steps and Stages in the Dental Laboratory

Phase 1—Clinical

Preliminary impression taken using irreversible hydrocolloid (Alginate; Kromopan LASCOD, Sesto Fiorentino, Italy) for provisional acrylic resin element fabrication.Abutment preparation: 6% axial wall taper, 1.5–2 mm occlusal reduction and deep chamfer finishing line.Temporary relining and cementation of the preliminary provisional prosthetic framework with eugenol-free Zinc Oxide cement (Temp-Bond™; Orange, CA, USA).After 30 days, the definitive impression made by mechanical retraction of the marginal gingival tissues using 100% cotton non-impregnated cord (Ultrapak™ Clean Cut; Ultradent™, Corsico, Italy), then, precision by single phase bicomponent technique using both high-viscosity (3M™ Impregum™ Penta™ H DuoSoft™; 3M, Saint Paul, MN, USA) and low-viscosity (3M™ Impregum™ Garant™ L DuoSoft™; 3M, Saint Paul, MN, USA) polyether-based impression material.

Phase 2—Dental Laboratory

5.Extra hard plaster model development—4th type (Fujirock Ep Classic; GC Corporation, Tokyo, Japan).6.Model scanning by 3Shape D500 laboratory scanner.7.CAD design and CAM fabrication of a resin prototype.8.Resin prototype placement on the abutment for functional and morphological evaluation and color definition.9.Fabrication of the final prosthetic restoration with Monolithic Zirconia Biodynamic Zirconium Multilayer 1200/600 Mpa Progressive (Biodynamic, Correggio, Italy). This is a Class IIa device approved by the Italian Health Ministry.10.Clinical evaluation of morphological and marginal accuracy (Fit Checker™ Advanced Blue; GC Corporation, Europe A.G. 2020).11.Final cementation of the polished, stained and glazed definitive crowns using self-adhesive universal composite cement (RelyX™ Unicem Aplicap™; 3M, Saint Paul, MN, USA).12.Follow-up visit 1 week later, then at 1 month and subsequently annually.

### 2.3. Systematic Clinical Evaluation

The clinical evaluation was carried out by two experienced prosthodontists using standardized parameters of the California Dental Association (CDA.), as widely accepted by the international scientific community [14]. Rating results regarding color, surface, anatomical shape and marginal integrity of the crown were described with a letter, as follows: R for Excellence, S for acceptability, T for Not Satisfying and V for Failure.

The CDA parameters are listed in Table 1.

Each parameter has a numerical correspondence:Surface;Color;Anatomical Shape;Marginal Integrity.

This method allows a detailed description and comparative assessment of the prosthesis’ qualitative status through the observational period, using acronyms composed of the alphanumeric series referring to the four parameters and their level of maintenance. The authors divided all CDA categories into 3 (a, b, c) to simplify data collection and interpretation. The evaluation criteria for scoring are reported in Table 2.

An overall score was given to the crowns as follows:Excellent: All parameters were consistent with the letter a;Acceptable: At least one clinical parameter corresponded to the letter b;To be re-done: At least one clinical parameter corresponded to the letter c.

Authors considered only the excellent cases as successes, meanwhile the excellent and acceptable cases were allowed to remain in place.

### 2.4. Follow-Up Program

After prosthesis placement, the first follow-up visits were performed 1 week and 1 month later, followed by an annual follow-up for a period of 5 years. Patients who did not complete all annual visits were excluded from the study.

Two experienced clinicians conducted the follow-up visits with a re-evaluation of the CDA parameters. These were collected in an appropriate Excel file for subsequent data analysis.

### 2.5. Statistical Analysis

A statistical analysis was performed to assess the correlation between failures and possible causes with a univariate logistic regression, considering the null hypothesis the lack of correlation between failure and other independent variables.

Kaplan–Meier survival method was used to plot the prosthesis success. All statistical tests were two-tailed, and the significance level was set at *p* < 0.05. All analyses were performed using Stata^®^, version 13.0 (Stata Corp., College Station, TX, USA).

## 3. Results

### 3.1. Description of the Examined Sample

Thirty-one patients were enrolled according to the inclusion/exclusion criteria. Fifteen were males (48.4%) with an age range of 25–82 years old (mean 63 years old) and sixteen females (51.4%) with an age range of 40–79 years old (mean 55 years old) with a total sample average age of 59.3 years. Of the 31 patients examined, 7 underwent prosthetic rehabilitation on more than one tooth, ranging from 2 to 3 prosthetic elements, while 24 patients underwent single crown rehabilitation.

Prosthetic rehabilitation was in the upper dental arch in 15 patients (48.4%) and in the lower dental arch in the remaining 16 (51.6%). Crowns represented 40 cases: first premolars were involved in 2 cases (5%), second premolars in 18 (45%), first molars in 15 (37.5%), and second molars in 5 (12.5%). Thirty-seven teeth (92.5%) had already been treated endodontically, of which sixteen (40%) had also been treated with endodontic posts. The opposite occlusal surface (47.50%) was artificial in 19 cases and natural in 21 cases (52.5%). These data are collected in Table 3.

### 3.2. Results Evaluation According to the C.D.A. Criteria

During the first and second year, all crowns maintained the delivery score stably; accordingly, 38 were “excellent”, while 2crowns (patients 8 and 26) presented slight color mismatch (1a, 2b, 3a, 4a) (Figure 1).

At the third-year evaluation, two patients showed changes compared to the initial evaluation, though they presented a survival rate of 100% and a success rate of 90% (Figure 2). More precisely:Patient 23 demonstrated increased surface roughness, resolved by intra-oral finishing (1a/2b/3a/4a).Patient 10 demonstrated a slight marginal gap on the vestibular aspect of the 36 tooth (1a/2a/3a/4b).

In the fourth year of evaluation, 34 patients maintained the same score of “excellent”, while 2 showed changes. More precisely (Figure 3):Patient 24 showed the first failure consisting of crown fracture, thus needing a replacement (1a/2a/3a/4c)Patient 12 showed slight surface roughness (1b/2a/3a/4a)

At the 5-year follow-up, 34 patients were still stable with an “excellent” score; 1 patient, already showing slight alteration at the third year, became the second case “to be re-done” (Figure 4).

Patient 10, who had shown slight marginal changes at the third year evaluation, at this stage showed an exposure of dentin (TMD) or base (TMB) at the marginal level (1a/2a/3a/4c) at the vestibular surface of the 36 tooth with secondary caries, leading clinicians to replace the crown.

In the fifth year of observation, the survival rate was 95%. More precisely, 34 cases (85%) were “Excellent” (1a/2a/3a/4a), 4 cases (10%) were “Acceptable” (at least one “b” in the rating), there were 2 cases of failure (5%) that occurred (at least one “c”) with a “To Be Re-done” rating, resulting in an 85% success rate (Figure 5).

Regarding the alteration analysis compared to the initial condition, the “not excellent” cases showed alteration “concerns” regarding “color” in two instances (33.3% of variated parameters, 2.5% of total), “surface” in two (33.3% of variated parameters), “anatomical shape” in one (16.7% of variated parameters) and “marginal integrity” in one (16.7% of variated parameters, 5% of total) (Figure 6).

### 3.3. Statistical Analysis

The result of the logistic regression showed no statistically significant relationship between failure occurrence and sex, age, prosthetic extension, previous endodontic treatment, presence of endodontic posts, type of antagonist and tooth position. The results are listed in Table 4.

The Kaplan–Meier curves concerning success and survival are separately reported in Figure 7a,b.

## 4. Discussion

The main objective of the current study was to evaluate the survival and clinical outcomes of Monolithic Zirconia on natural abutments in the posterior sectors performed by less-experienced clinicians (dentistry students in the final year of bachelor study) to potentially determine if this kind of dental rehabilitation may be considered a dependable treatment within everyone’s reach.

Multilayer 5Y-TPZ blank was selected as the more appropriate material for the final restoration because of its good optical properties, less need for tooth preparation [15] and to minimize the possibility of error for students. Considering that inexperienced practitioners may also have some difficulties in the correct evaluation of the correct interocclusal reduction, authors set a minimum of 1.5 mm as the interocclusal space to prevent prosthetic crown fracture. This value was decided on as it is generally considered that 0.6–0.8 mm of interocclusal reduction may be associated with a high fracture risk [16,17].

Among our 40 crowns, a single framework fracture occurred after 5 years of observation. More precisely, the fracture involved the connector area of a prosthetic pontic involving 4.5 and 4.7 teeth, which had a prosthetic antagonist too. Several studies have deepened these issues, converging on the conclusion that although Zirconia demonstrates excellent hardness values, in the case of rehabilitation that include pontic, a large connector surface area is mandatory to reduce the risk of fracture [18,19].

During the study, significant attention was paid to the preparation taper; indeed, the. tutors helped students, when necessary, to restore the correct abutment shape when preparation of axial walls was more than 6°, in order to obtain the best taper in each case. It is probable that, for this reason, no de-cementation was observed among our cases. These data contrast with other reports in which de-cementation has been identified as the most frequent technical problem [20]. The cementation material could also affect this result; the authors decided to use resin self-adhesive composite cement for its greater percentage of success rather than conventional zinc-phosphate and glass ionomer cement [21].

In line with Rinke et al., the authors considered that the high marginal precision offered by the latest generations of CAD-CAM technologies allows clinicians to perform better [22]; therefore, deep chamfer margin preparation was chosen because it is simpler than a 90° shoulder or juxta-gingival chamfer, is more aesthetic and gives restoration a correct geometry that reduces occlusal stress [23,24,25,26].

After a brief review of the relevant literature, we selected 13 clinical studies on Monolithic Zirconia single and multiple fixed prosthesis on natural abutments within 5 years of follow-up with a mean of 3.4 years follow-up. The authors compared these studies with their results of year-by-year survivals and CDA rating. In the current study, no complications occurred during the first year of evaluation, with a 100% survival rate, the latter is consistent with different authors [24,25,26,27,28,29,30]; however, in contrast, Miura, Gunge and Bankoğlu Güngör reported initial failures after the first year, with abutment root fracture being the most common biological complication [17,31,32]. The overall survival rate in the second year was similar to ours, while different from other studies reporting two root fractures and one root fracture of the antagonist tooth [26,31,33].

Only during the third year did we observe mild alterations such as increased surface roughness and a slight marginal gap; however, our survival rate remained higher than other authors reporting prosthetic unrepairable framework fractures and root fractures in the same period [17,25,29].

In the fourth year, the first failures occurred among our patients. These consisted of FDP fracture between 45 and 46 and increased surface roughness. Similar results have been reported by Heller in his study, as three patients showed secondary caries after the fourth year, thus requiring crown removal, conservation treatment and crown replacement [28]. Among our cases, a similar complication occurred after five years. The same patient had already showed a slight marginal under-contouring in the third year evaluation. Secondary caries is undoubtedly the most frequent biological complication, as reported in several systematic reviews and meta-analyses [2,34]; of these, four studies evaluated the clinical outcomes of Monolithic Zirconia on natural posterior teeth by CDA parameters, all with a two-year follow-up, with the exclusion of Heller’s study with an 8-year follow-up. At the short-term of 2 years follow-up, our results regarding anatomy and color parameter (95% excellent) are certainly better when compared to the 82% and 90.2% of Levartovsky and Gseibat, respectively, and very similar to the 95.9% reported by Tang et al. [26,27,33].

Surface roughness and marginal integrity were the parameters that varied from excellent to acceptable (one case [2.5%]) during the observational period in our study; similar results have been previously reported by Gseibat, Tang and Levartovsky [26,27,33]. We observed surface roughness increasing in two patients receiving a FDP. This may be explained by the Low-Temperature Aging effect on Monolithic Zirconia after occlusal adjusting performed intra-orally by diamond burst [34,35].

Overall, the survival rate of 95% we reported is similar to several authors in the third year of follow-up [24,26,27,33], consistent with the results of Solà Ruiz, Waldecker and Heller in the fifth year [25,28,36], and better than Miura, Gunge, Bankoğlu Güngör and Mikeli [17,29,31,32].

A similar success rate of 90% at 2- and 3-year follow-ups of the “success” cases (“excellent” CDA parameters) has been previously reported [27,33,37]; additionally, our 85% success rate at 5 years is comparable to Heller’s data, who reported a success rate of 73.4% at 8 years follow-up [28].

After the statistical analysis, the null hypothesis was accepted: complications that occurred within 5 years of follow-up are unlikely to be statistically related to any characteristic of the prosthetic restoration considered in this sample.

Our study success and survival rates are consistent with interesting studies previously published that evaluated the outcomes of Zirconia-layered restorations made by predoctoral dental students. Philaia et al. evaluated different types of Veneered Zirconia (Zirkonzahn Zirconia, NobelProcera Zirconia, and Prettau Zirconia), reporting 89% success and 100% survival rates in an evaluation period between 3 and 7 years, despite showing technical complications (chipping or porcelain fracture) [12]. Näpänkangas et al. evaluated the same types of Veneered Zirconia, leading to a similar result: a success rate of 80% and a survival rate of 89% at 4 years of follow-up [13]. Both concluded that Zirconia restorations are suitable for dental students.

## 5. Conclusions

Although a worldwide experience of the long-lasting duration already exists on the use of Monolithic Zirconia for single or multiple tooth restoration, the authors would like to stress its use by less experienced clinicians (final-year undergraduate students) with such restorations representing valid alternatives to the conventional metal–ceramic, which requires different and more accurate handling. This study suggests that under the supervision of experienced tutors, dental students could also reach excellent results for similar restorations, showing Monolithic Zirconia rehabilitation to be less operator-dependent and having a shallow learning curve.

Several limitations must be considered for the current study, such as the small size of the sample, the lack of evaluation of periodontal and plaque index and the lack of a randomized clinical trial or split mouth analysis. Further studies on larger samples and with longer follow-up, possibly also considering additional parameters, are needed to further support Monolithic Zirconia use as the gold standard for fixed rehabilitation of the posterior sectors. 

### Clinical Case

The clinical case iconography is reported in the figure below (Figure 8).

## Figures and Tables

**Figure 1 ijerph-20-02943-f001:**
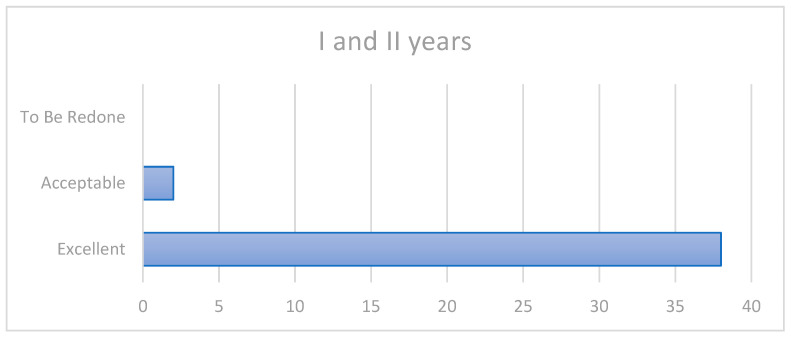
First and second years’ outcomes.

**Figure 2 ijerph-20-02943-f002:**
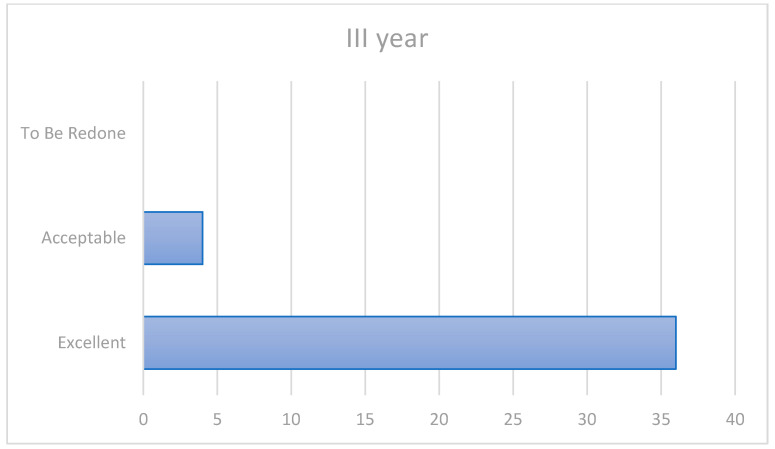
Third-year outcomes.

**Figure 3 ijerph-20-02943-f003:**
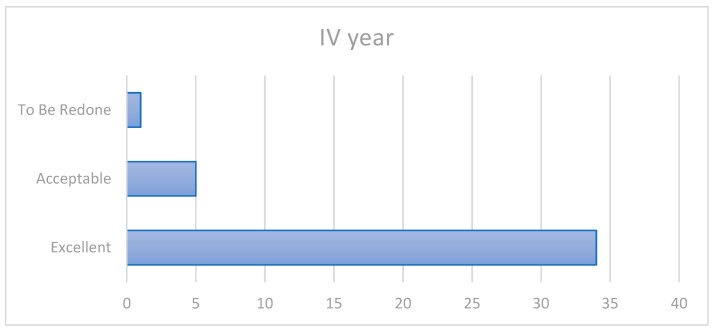
Fourth-year outcomes.

**Figure 4 ijerph-20-02943-f004:**
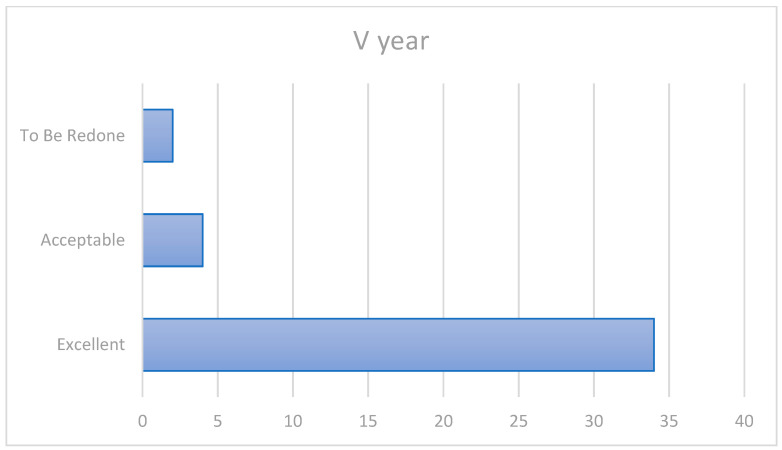
Fifth-year outcomes.

**Figure 5 ijerph-20-02943-f005:**
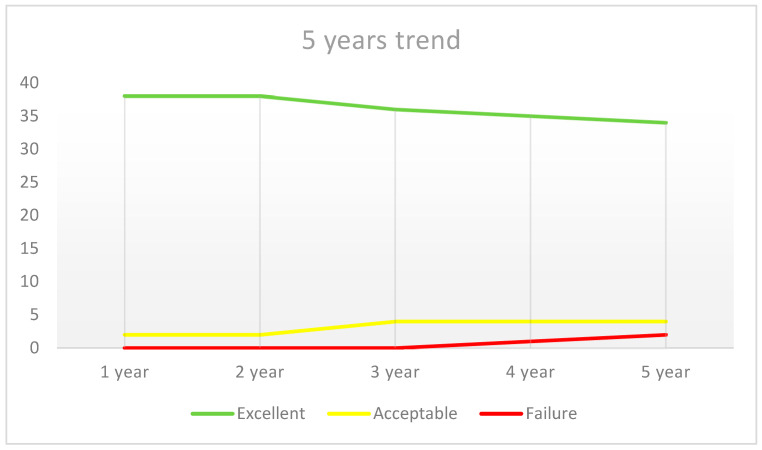
Five-year outcomes trend.

**Figure 6 ijerph-20-02943-f006:**
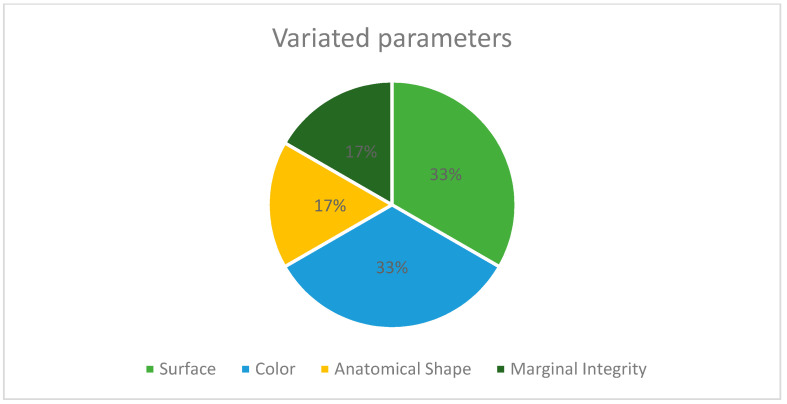
Distribution of parameters changed during the observational period.

**Figure 7 ijerph-20-02943-f007:**
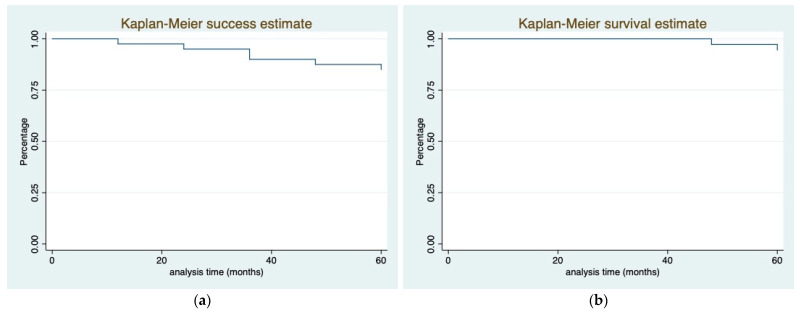
Kaplan–Meier curves plot. (**a**) success plot; (**b**) survival plot.

**Figure 8 ijerph-20-02943-f008:**
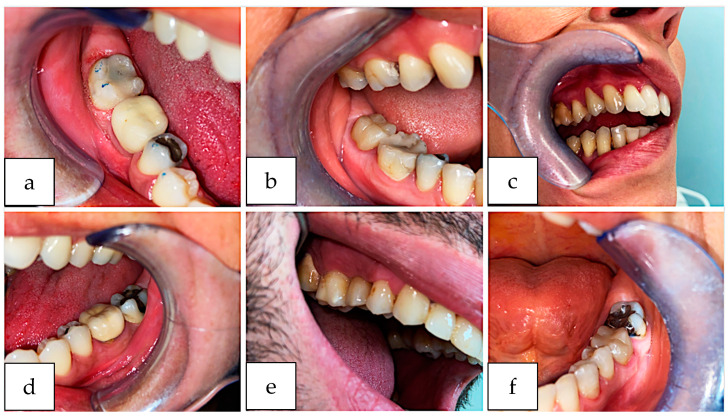
(**a**) Slight color mismatch on 46 crown; (**b**) Monolithic Zirconia single crown on 47; (**c**) Monolithic Zirconia single crown on 24; (**d**) Monolithic Zirconia single crown on 36; (**e**) Monolithic Zirconia single crown on 15; (**f**) Monolithic Zirconia single crown on 36.

**Table 1 ijerph-20-02943-t001:** Rating standardized parameters of the California Dental Association.

CDA Rating	Abbreviated CDA. Code	Meaning
Excellence	R	Range of excellence
Satisfying	S	Range of acceptability
Not satisfying	T	Replace or correct restoration for prevention
Failure	V	Restoration failure: must be replaced

**Table 2 ijerph-20-02943-t002:** CDA rating parameters definition.

	Rating (CDA)	Definition
1-Surface		
a	R0	Absolutely perfect
	R1	The surface of the restoration is smooth. No irritation of adjacent tissues
b	S	The restoration surface is slightly rough or porous; it can be polished
c	T	Deeply porous surface; irregular grooves; cannot be polished
	V	Fractured or flaking surface
2-Color		
a	R0	Absolutely perfect
	R1	Slight discrepancy in color, shade or translucency
b	S	Discrepancy between restoration and tooth structure with a normal range of color
c	T	Discrepancy between restoration and tooth structure outside the normal range of color
	V	Aesthetically unpleasing tooth color, shade or translucency
3-Anatomical Shape		
a	R0	Absolutely perfect
b	S	The restoration is slightly under-contoured Interproximal contact is slightly open The restoration is slightly over-contoured but can be reduced
c	T	The restoration is under-contoured: dentin or base is exposed Occlusion is not correct: the contact is faulty Interproximal contact is open: probable tissue damage. The restoration is over-contoured, but it cannot be appropriately adjusted
	V	Incongruous or lost restoration Occlusion is traumatic
4-Marginal Integrity		
a	R0	Absolutely perfect
	R1	The periodontal probe meets no obstacles, but differences in height are heard
b	S	Grooves are found along the margins, which do not extend beyond the amelo-dentinal junction. Slight margin discoloration
c	T	Marginal abutment exposition, from point-like to wide exposition
	V	The restoration or tooth structure is fractured Continuous caries are evident on the restoration margin

**Table 3 ijerph-20-02943-t003:** Sample description.

Patient	Sex	Age	Prosthetic Crowns	Tooth	Endodontic Treatment	Number of Endodontic Posts	Antagonist Surfaces
1	M	70	1	45	1	0	Artificial
2	F	40	1	25	1	1	Natural
3	F	56	1	15	1	1	Artificial
4	F	59	1	35	0	0	Artificial
5	F	40	1	16	1	0	Natural
6	F	49	2	25–26	2	2	Artificial
7	M	67	1	15	1	1	Artificial
8	M	59	1	37	0	0	Artificial
9	M	73	1	45	1	1	Natural
10	M	25	1	36	1	0	Natural
11	F	42	1	16	1	0	Natural
12	M	72	3	14–16	3	2	Artificial
13	M	55	2	36–37	2	1	Natural
14	F	62	1	45	1	0	Artificial
15	F	56	2	36–37	2	1	Artificial
16	F	72	1	35	1	0	Artificial
17	M	82	1	35	1	1	Natural
18	F	70	1	14	1	0	Artificial
19	F	74	1	35	1	1	Artificial
20	F	45	1	26	0	0	Artificial
21	M	51	1	26	1	0	Natural
22	M	66	2	25–26	2	2	Natural
23	M	57	2	15–16	2	1	Natural
24	M	53	3	45–47	3	1	Artificial
25	M	53	1	25	1	0	Artificial
26	F	69	1	46	1	1	Natural
27	M	78	1	35	1	0	Artificial
28	F	72	1	15	1	1	Artificial
29	F	45	1	36	1	0	Natural
30	M	79	1	17	1	1	Artificial
31	F	65	1	46	1	0	Artificial

**Table 4 ijerph-20-02943-t004:** Results of the univariate logistic regression between failures and different variables (statistical significance for *p* < 0.05).

				*p* < 0.05
Sex	M	F		0.360
	15	16		
Age	Minimum	Maximum	Mean	0.506
	25	82	59.3	
N. of Prosthetic Rehabilitation	1	2	3	0.995
	24	5	2	
Endodontic Treatment	Yes	No		0.996
	37	3		
Presence of Fiber Post	Yes	No		0.737
	16	24		
Opposite Tooth Surface	Natural	Artificial		0.864
	21	19		
Location (Patients)	Maxillary	Mandibular		
	15	16		0.685
Type of Tooth	Molar	Premolar		
	20	20		0.450

## Data Availability

Not applicable.

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
