# Peer review of "Clinical Outcomes of Monolithic Zirconia Crowns on Posterior Natural Abutments Performed by Final Year Dental Medicine Students: A Prospective Study with a 5-Year Follow-Up"

_ijerph, 2023, doi:10.3390/ijerph20042943_

Round 1
Reviewer 1 Report
I would try to update part of the refernece list and if possile, I would suggest to dimish the number of the quted article
better perfromed images might be helpful for the reader and might turn on a spot light on the excellent work done by the authors
limitation of the protocol should be better highlighted in the discussion session
1. Main question is to test the viability of a full zirconia crown
2. Its not a novelty but the long term follow up might be of interest for the reader 3. The long term follow up, as mentioned before, is the key 4. Methodology is difficult to improve because the nature of the trial (in human clinical study). What can be added is the periodontal parameters 5. The conclusions consistent with the evidence and argumentspresented and they address the main question posed 6. References are surprisingly correct. They might be updated sometimes, but there is not any self reference. 7. Tables are ok, figures might be taken (or post-produced) correctly
Author Response
Thank your very much for your suggestions. The paper has been modified where required according to the relevant suggestions and overall improved by adding what kindly evaluated.
I would try to update part of the refernece list and if possile, I would suggest to dimish the number of the quted article
- Thank you for this suggestion. Reference list has been modified keeping the latest literature articles. 20 references were removed.
better perfromed images might be helpful for the reader and might turn on a spot light on the excellent work done by the authors
- Thank you, images were post-produced increasing contrast and sharpness.
limitation of the protocol should be better highlighted in the discussion session
- Thank you: section regarding limitations were improved accordingly.
Main question is to test the viability of a full zirconia crown
- Corrected into the main text
- Its not a novelty but the long term follow up might be of interest for the reader
- Added in limitations section as suggested
- The long term follow up, as mentioned before, is the key
- Added in limitation section
- Methodology is difficult to improve because the nature of the trial (in human clinical study). What can be added is the periodontal parameters
- Added in limitation sections and furthers study
- The conclusions consistent with the evidence and arguments presented and they address the main question posed
- Thank you
- References are surprisingly correct. They might be updated sometimes, but there is not any self reference.
- References list was updated. Thank you
- Tables are ok, figures might be taken (or post-produced) correctly
- Figures were post-produced, thank you.
Authors want to truly thank the reviewer 1 for considering our study, understanding the limitations occurred, especially in human trial. These suggestions and comments were very useful to improve the manuscript in order to be clearer to reader. Thank you.
Reviewer 2 Report
Manuscript title
The Attitude Of Dental Medicine Students In Prosthetic Rehabilitation Of Posterior Natural Abutments By Monolithic Zirconia: A Perspective 5-Year Follow-Up Study Based On California Dental Association Parameters
The authors present results from a study conducted in Italy to evaluate the 5-year clinical outcomes of Monolithic Zirconia prosthetic crowns on natural abutments in the posterior sectors, performed by final year dental students (as so considered as less experienced clinicians). The manuscript is a bit hard to follow. This study could be of interest, but requires changes prior to publication.
Abstract
- In general, the abstract is not well balanced. According to MDPI policies, the abstract is not structured and must be presented in a single paragraph.
- Too much information presented from the background before of the objective.
- The objective of the summary and the text do not match, consider unifying it.
- The authors did not state the study design. It is necessary to add it.
- Mention the setting, city and country where the study was conducted.
- Consider mentioning the age range of the participants, % sex (Describe the sample).
- Did the authors perform any type of data analysis? (basic analysis?)
- Present the results for (average) age in the results and sex (percentage).
- Reconsider rewriting the conclusion based on the current study and research results
Introduction
- Consider the introduction classic 3-paragraph: What we know; what we no-know; and why this study was done. Please consider splitting the single paragraph of the introduction to make it more understandable to the reader.
- What is the research question and what is the hypothesis?
- Explain the reasons and the scientific basis of the investigation. What is the rationale for the study?
- The authors do not mention any antecedents about attitudes that they mention in the title. In the introduction they mention the physical qualities of zirconia. In general, they do not introduce the problem that they should try to solve in the study.
Material and Methods
In general, the methodology needs to be described in greater detail to be accepted. What do the authors intend to present? What study design did you do? I believe that the authors should consult what should be reported according to the study carried out and resubmit their study.
- The authors did not state the study design. Please define.
- Mention the setting, city and country where the study was conducted.
- Mention the country where the study was conducted.
- How did the authors arrive at the sample? Was there any sample size calculation?
Results and
- This section is very difficult to follow. It does not appear to be a scientific article or the authors fail to convey what they want to say. The authors should make an improvement in the presentation of the results.
- what is the usefulness of the manuscript?
- In the results the authors mention that they used logistic regression, however in the material and methods section they do not mention anything about it. On the other hand, the sample size does not allow for an adequate logistic regression analysis.
Discussion
The authors present the discussion in a single paragraph. Ideas need to be better structured.
Conclusion
- Reconsider rewriting the conclusion based on the current study and research results. Please elaborate.
Author Response
Thank your very much for your suggestions. The paper has been modified where required according to the relevant suggestions and overall improved by adding what kindly evaluated
Abstract
- In general, the abstract is not well balanced. According to MDPI policies, the abstract is not structured and must be presented in a single paragraph.
- Too much information presented from the background before of the objective.
- The objective of the summary and the text do not match, consider unifying it.
- The authors did not state the study design. It is necessary to add it.
- Mention the setting, city and country where the study was conducted.
- Consider mentioning the age range of the participants, % sex (Describe the sample).
- Did the authors perform any type of data analysis? (basic analysis?)
- Present the results for (average) age in the results and sex (percentage).
- Reconsider rewriting the conclusion based on the current study and research results
Thank you very much for suggestions; the abstract section has been modified following point-by-point suggestions
Introduction
- Consider the introduction classic 3-paragraph: What we know; what we no-know; and why this study was done. Please consider splitting the single paragraph of the introduction to make it more understandable to the reader.
- Introduction section has been modified/corrected accordingly
- What is the research question and what is the hypothesis?
- The research question was to evaluate the clinical outcomes of Monolithic Zirconia performed by students while the hypothesis is that Monolithic Zirconia restorations are predictably regardless clinician experience. As so such suggestion was addressed in introduction.
- Explain the reasons and the scientific basis of the investigation. What is the rationale for the study?
- The rationale was to generally investigate about the possibility for dental school students (so conceptually with less experience) to manage rehabilitation with a widely used material nowadays.
- The authors do not mention any antecedents about attitudes that they mention in the title. In the introduction they mention the physical qualities of zirconia. In general, they do not introduce the problem that they should try to solve in the study.
- Thank you for this comment. It was specified in introduction section.
Material and Methods
In general, the methodology needs to be described in greater detail to be accepted. What do the authors intend to present? What study design did you do? I believe that the authors should consult what should be reported according to the study carried out and resubmit their study.
- The authors did not state the study design. Please define.
- Defined better in the beginning of Material and Methods. Thank you.
- Mention the setting, city and country where the study was conducted.
- Defined better in the beginning of Material and Methods. Thank you.
- Mention the country where the study was conducted.
- Defined better in the beginning of Material and Methods. Thank you.
- How did the authors arrive at the sample? Was there any sample size calculation?
- As for the ethical committee guidelines, patients were enrolled just in one year and so 31 were collected for the current study
Results and
- This section is very difficult to follow. It does not appear to be a scientific article or the authors fail to convey what they want to say. The authors should make an improvement in the presentation of the results.
- Result section has been improved to better explain what collected data leaded to.
- what is the usefulness of the manuscript?
- To provide to the scientific community further data about a current widely used material in dental prosthesis.
- In the results the authors mention that they used logistic regression, however in the material and methods section they do not mention anything about it. On the other hand, the sample size does not allow for an adequate logistic regression analysis.
- The last section of materials and methods explains the statistical analysis adopted to evaluate some statistical differences between patients and prosthetic characteristics and failures although on a limited number of cases.
Discussion
The authors present the discussion in a single paragraph. Ideas need to be better structured.
- The section has been modified according to improve the readability of the paper.
Conclusion
- Reconsider rewriting the conclusion based on the current study and research results. Please elaborate.
- Conclusion section has been modify accordingly
Reviewer 3 Report
First of all, I would like to greet the authors and congratulate them on the theme and work done. The study appears correctly performed and written without logical or factual errors. The tables and graphics presented are very clear and facilitate the understanding of Methodology and Results.
Authors have well revised several issues. The following comments are addressed and require minor modifications to enhance the quality of the manuscript:
-I would begin with the title of the work which I kindly suggest being reviewed by the authors. The term “The attitude”, in my opinion, does not fit the development of the theme, because students were limited to the realization of fixed rehabilitation, and the follow-up was carried out by two experienced prosthodontists. (I suggest something like” Clinical outcome of prosthetic rehabilitation of posterior…performed by Dental Medicine students: a 5 year ….”) On the other hand, I would like to be clarified by the term perspective study, since the literature mentions prospective and retrospective observational studies.
-In abstract line 30 you mention “each prosthetic restoration was defined by the terms Excellent, Acceptable an To be re-done” which do not match the terms in table 1 (CDA rating).
-Line 74 must be revised.
-Line 311 e 312-word chamfer
-Line 362 the null hypothesis is mentioned by the first time, I suggest to be mentioned after or before objectives, in introduction.
-The photos on page 14 (figure 7) are clear but should be cropped to see the details.
Author Response
Thank your very much for your suggestions. The paper has been modified where required according to the relevant suggestions and overall improved by adding what kindly evaluated
First of all, I would like to greet the authors and congratulate them on the theme and work done. The study appears correctly performed and written without logical or factual errors. The tables and graphics presented are very clear and facilitate the understanding of Methodology and Results.
Authors have well revised several issues. The following comments are addressed and require minor modifications to enhance the quality of the manuscript:
-I would begin with the title of the work which I kindly suggest being reviewed by the authors. The term “The attitude”, in my opinion, does not fit the development of the theme, because students were limited to the realization of fixed rehabilitation, and the follow-up was carried out by two experienced prosthodontists. (I suggest something like” Clinical outcome of prosthetic rehabilitation of posterior…performed by Dental Medicine students: a 5 year ….”) On the other hand, I would like to be clarified by the term perspective study, since the literature mentions prospective and retrospective observational studies.
- Thank you for your comment. It was addressed changing the title following your suggestion.
-In abstract line 30 you mention “each prosthetic restoration was defined by the terms Excellent, Acceptable an To be re-done” which do not match the terms in table 1 (CDA rating).
- Thank you. Abstract was modified
-Line 74 must be revised.
- This period was revised, in order to be clearer. Thank you
-Line 311 e 312-word chamfer
- Chamfer word revised. Thank you
-Line 362 the null hypothesis is mentioned by the first time, I suggest to be mentioned after or before objectives, in introduction.
- The null hypothesis definition was added in introduction and material and methods. Thank you.
-The photos on page 14 (figure 7) are clear but should be cropped to see the details.
- Images were post-produced by increasing contrast and sharpness. Thank you for suggestion
Authors would sincerely thank the Reviewer 3 for kind words about our manuscript. Your suggestions were very meaningful and useful in order to improve the readability. Thank you very much.
Reviewer 4 Report
It is an interesting clinicl work about ZrO crowns for nature dentlal abutment. The results showed the ZrO is an excellent material for dental crown. The topic of this work is suitable for this journal.
But the authors need to be noted this problem.
Why only 40 cases were slected for this study? The cases for study seem not enough for the analysis.
I think the main problem for this paper is that the data from 40 ZrO crown for 31 patients is not enough for the analysis. I am not sure if the authors has more case data about this study. If they have, the conclusion for this paper might be more reasonable.
Author Response
Thank your very much for your suggestions. The paper has been modified where required according to the relevant suggestions and overall improved by adding what kindly evaluated
It is an interesting clinicl work about ZrO crowns for nature dentlal abutment. The results showed the ZrO is an excellent material for dental crown. The topic of this work is suitable for this journal.
But the authors need to be noted this problem.
Why only 40 cases were slected for this study? The cases for study seem not enough for the analysis.
I think the main problem for this paper is that the data from 40 ZrO crown for 31 patients is not enough for the analysis. I am not sure if the authors has more case data about this study. If they have, the conclusion for this paper might be more reasonable.
- Authors wants to thank you for appreciation words about our work. Unfortunately, the sample size was forced by local ethical committee, which gave authors permission to carry out this study within 1 year. These 31 patients were afferent to University Hospital Policlinic of Bari during 2017 (added in materials and methods). Authors, to be rigorous, considered only the patients that fit in the inclusion criteria in 2017, in order to have 5 years of follow up, ended in 2022. Sample size is one of main limitations of this study, as indicated in limitations section and conclusion suggests that further study with larger samples must be conducted. Moreover, authors acknowledge that carry out clinical study along many years of follow up present some difficulties, firstly the adherence of patients to follow up visits. But larger sample may consist in more power study with stronger scientific soundness. Authors regret about small sample size but was indirectly imposed by local ethical committee rules. Thank you for such specification.
Round 2
Reviewer 2 Report
The authors have made the changes proposed by me properly.
Author Response
Dear reviewer 2,
thank you very much for the suggestions that you had proposed that have substantially improved paper quality.
Best regards